# Collateral assessment on magnetic resonance imaging/angiography up to 30 hours after stroke onset

Shinya Tomari[1]*, Thomas Lillicrap[1], Carlos Garcia-Esperon[1,2,3], Yumi Tomari Kashida[1], Andrew Bivard[4], Longting Lin[1], Christopher R. Levi[1,2,3], Neil J. Spratt[1,2,3]

1 Hunter Medical Research Institute, Newcastle, Australia, 2 Department of Neurology, John Hunter Hospital, Newcastle, Australia, 3 College of Health, Medicine, and Wellbeing, University of Newcastle, Newcastle, Australia, 4 Melbourne Brain Center at the Royal Melbourne Hospital, University of Melbourne, Parkville, Australia

* sny5588@gmail.com

## Abstract

### Purpose

We aimed to validate hyperintense vessel sign (HVS) on FLAIR imaging or posterior cerebral artery (PCA) laterality on MR angiography beyond 4.5 hours after stroke onset.

### Materials and methods

Data from acute ischemic stroke patients with internal carotid or middle cerebral artery occlusion who underwent CT perfusion imaging at baseline, follow-up MR perfusion imaging and angiography within 30 hours after stroke, without effective recanalization on follow-up imaging, were analysed retrospectively. Patients were separately classified as high or low HVS (>5 or $\leq$5 slices of HVS), and PCA laterality positive or negative group. We compared core and penumbra volumes at follow-up imaging and neurological outcomes between high or low HVS group, and between PCA laterality positive or negative group.

### Results

Of 49 patients analyzed, four patients with artifacts were excluded and 45 were classified into high (n = 23) or low (n = 22) HVS group. High group had a smaller core volume (median 32 ml versus 109 ml, p = 0.004), larger penumbra volume at follow-up (68 ml versus 0 ml, p = 0.001), and better outcomes (modified Rankin Scale at three months, 3 versus 5, p = 0.03). For PCA laterality analysis, 1 patient with previously occluded PCA was excluded and 48 patients were classified as positive (n = 22) or negative (n = 26). Positive group had larger core volume (116 ml versus 37 ml), and no significant differences in penumbral volumes or outcomes.

### Conclusion

Prominent HVS in later time was associated with small core volume, persistent penumbra volume and favorable outcomes.

**Data Availability Statement:** All relevant data are within the manuscript and its Supporting Information files.

**Funding:** The author(s) received no specific funding for this work.

**Competing interests:** The authors have declared that no competing interests exist.

## Introduction

The time window for thrombectomy extends up to 16–24 hours from symptom onset for ischemic stroke patients with persistent target mismatch [1, 2]. Patients with good collateral status are likely to have persistent target mismatch and previous studies reported collateral status grading based on computed tomography (CT) angiography [3, 4] or continuous parameters calculated with perfusion imaging such as hypoperfusion intensity ratio [5] or CT perfusion collateral index (CTPCI) [6]. Collateral assessment on magnetic resonance imaging (MRI) such as hyperintense vessel sign (HVS) on fluid-attenuated inversion recovery (FLAIR) [7, 8] or posterior cerebral artery (PCA) laterality on magnetic resonance angiography (MRA) [9] has been previously reported to be associated with good collateral supply.

HVS represents retrograde blood flow through the leptomeningeal anastomosis (LMA) during acute ischemic stroke [7, 8], and prominent HVS has been associated with good collateral flow, small core volume and favorable neurological outcomes [10, 11]. One or more vessel segments observable in PCA P4 segment beyond the extent of filling seen in the contralateral PCA represents the existence of collateral flow from the PCA via the LMA (PCA laterality) [9] and the presence of the laterality before thrombolysis predicts good recanalization and favorable outcomes [12]. Both signs were evaluated within 4.5 hours after stroke onset. Increasingly, patients presenting in later time windows are considered for EVT, or late-window thrombolysis, so it is important to know whether these MRI signs may be useful beyond 4.5 hours. We aimed 1) to identify acute ischemic stroke patients with HVS or PCA laterality between 4.5 hours and 30 hours after stroke onset, and 2) to compare core and penumbra volumes at follow-up imaging and neurological outcomes between those with or without prominent HVS and those with positive or negative PCA laterality.

## Methods

### Population and data collection

We retrospectively identified patients from our prospectively collected data base of acute ischemic stroke patients presenting to the John Hunter Hospital (New South Wales, Australia) from 1st June 2009 to 30th July 2017. Selected patients underwent acute CT perfusion imaging at baseline (within 6 hours after stroke onset or last seen well), with follow-up MR perfusion and angiography within 30 hours after stroke. We included patients who had acute large vessel occlusion (internal carotid artery (ICA) isolated occlusion or tandem occlusion, or MCA segment 1 or segment 2 proximal portion, without effective recanalization between baseline and follow-up imaging. Patients with M3 or distal occlusion or those with ICA/MCA occlusion successfully revascularized by thrombolysis or thrombectomy were excluded. Assessment of recanalization was made using the modified thrombolysis in cerebral infarction (mTICI) grade, with assignments of 0 (no recanalization), 1 (minimal recanalization), 2a (partial recanalization with less than 50%), 2b (partial recanalization with more than 50%) and 3 (complete recanalization). Effective recanalization was defined as mTICI 2b to 3. Patients with previously occluded PCA were excluded from PCA laterality assessment. We recorded clinical characteristics (age, sex, National Institute of Health Stroke Scale (NIHSS) at baseline and follow-up (assessed at 24–72 hours after onset), risk factors) for enrolled patients. The study was conducted in accordance with national guidelines and had institutional ethical approval as part of INSPIRE research (HNELHD HREC Reference No: 11/08/17/4.01). An opt-out consent process was followed. The data were accessed on the 30th October 2022 for research purposes. Authors had access to information that could identify individual participants during or after data collection.

## Neuroimaging

**Baseline CT.**   CT imaging was derived from 320-slice Aquilion ONE scanner (Canon medical systems, Otawara, Japan). Image scanning started 7 seconds after intravenous injection (40 ml, injected at 6ml/s) of non-ionic iodinated contrast (Ultravist 370; Bayer Health-Care, Berlin, Germany). It lasted for 60 seconds, acquiring 19 images per slice.

**Follow-up MR image.**   MR imaging was performed on a 1.5- Tesla scanner (Siemens Avanto, Erlangen, Germany) and included axial isotropic diffusion weighted image (DWI), FLAIR image, time of flight magnetic resonance angiography, and bolus-tracking perfusion-weighted imaging. Following a bolus of gadolinium contrast (Magnevist; Bayer HealthCare, Berlin, Germany) into the antecubital vein (0.2 mmol/kg, injected at speed of 5 ml/s). The scanning lasted for 60 seconds, resulting in 40 images per slice. A total 19 slices were obtained.

**Perfusion image analysis.**   CT perfusion and MR perfusion imaging was post-processed using the commercial software MIStar (Apollo Medical Imaging Technology, Melbourne, Australia) to generate automated core-penumbra maps. Penumbra was defined as tissue with a Delay Time (DT) >3 seconds and relative cerebral blood flow (CBF) $\geq$30% of the contralateral hemisphere, using either CT perfusion or MR perfusion imaging in MIStar [13]. DT is an index of time to the peak of the residual function, similar to Tmax. Ischemic core was defined as the tissue with a DT >3 seconds and a relative CBF <30% of the contralateral hemisphere on CTP, or an apparent diffusion coefficient threshold of <620×10$^{-6}$ mm$^2$/s on MR-DWI. All images were reprocessed and analysed using the same version of the MIStar software (Version 3.2 release 3.2.62.03, last update October 2019)

## HVS and PCA laterality assessment

We focused on standard 5-mm-thick FLAIR images in the axial/horizontal plane with an intersection gap of 2 mm. We defined HVS as linear hyperintensities relative to gray matter in the MCA-territory. Punctiform hypersignals were not regarded as HVS. The HVS were differentiated from subarachnoid hemorrhage or inflammatory processes. FLAIR images in the horizontal plane were analyzed from the first M1-MCA appearance to the 10[th] image. We adopted the same scoring system in the previous paper [11]. The score was based on a rostrocaudal extension of HVS. Absence of HVS on 1 slice was rated 0 point. As 10 images were analyzed, resulting HVS ranged from 0 to 10 (Fig 1A). Patients were classified as high HVS (>5 slices of HVS on FLAIR) or low ($\leq$5) group. Patients with one or more vessel segments observable in PCA P4 segment beyond the extent of filling seen in the contralateral PCA, were classified as PCA laterality positive (Fig 1B). Two readers (S.T.; stroke neurologist, >10 years of experience and Y.T.K.; neurosurgeon, >5 year-experience of stroke clinical practice) each graded the HVS and PCA laterality for each patient, independently and regraded by consensus if the grading differed.

We compared CTPCI at baseline imaging, core and penumbra volumes at baseline and follow-up between high or low HVS group, and between PCA laterality positive or negative group. Further, we investigated neurological outcomes (modified Rankin Scale (mRS) at three months after stroke onset) of each group. The CTPCI is a continuous variable with lower value representing better collateral flow [6] and predicts slow core growth [14] and persistent penumbra [15]. The following equation was used to define the index: CTPCI = DT>6 /DT>2 × 100

## Statistical analysis

Continuous variables are presented as median with interquartile range. Clinical characteristics were compared between high or low HVS group and with or without PCA laterality using

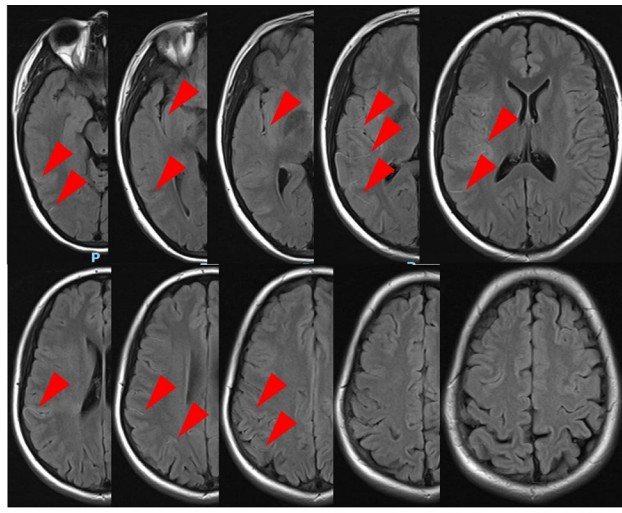

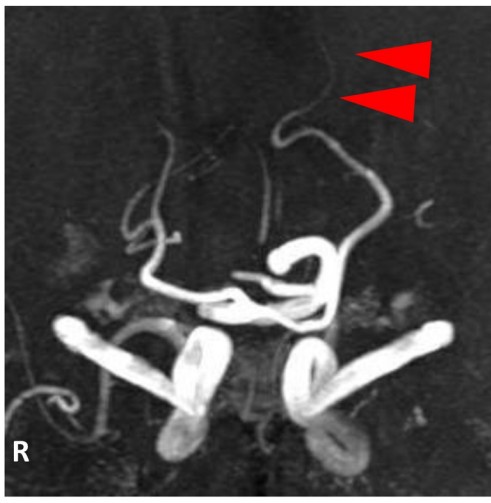

Standard 5-mm-thick FLAIR sections with an intersection gap of 2 mm

A. Hyperintense vessel sign on FLAIR image

B. PCA laterality on MR angiography

**Fig 1.** A: representative case presentation of HVS; 31 year-old female had right MCA M1 proximal occlusion. MRI FLAIR image was performed at five hours after stroke onset. The figure shows eight slices of HVS on FLAIR. The patient was classified as high HVS group. B: representative case presentation of PCA laterality; 77 year-old male had left MCA M1 proximal occlusion. MRA was performed at 19 hours after stroke onset. The figure shows observable extended PCA P4 segment.

Pearson's chi-square test for categorical variables, or Wilcoxon's rank sum test for continuous variables. Multiple regression analysis was performed with core volume at follow-up as a dependent variable, incorporating HVS, PCA laterality, occlusion site and thrombolysis as independent ones. All statistical analysis was done using STATA 15.0 (Stata Corp, College Station, Texas, USA), with significance level set at 0.05. The datasets generated for this study are available on request to the corresponding author.

## Results

We reviewed 272 patients who had baseline CT perfusion imaging within 6 hours after stroke onset and follow-up MR perfusion imaging/angiography within 30 hours after stroke (Fig 2). We excluded patients with effective recanalization between baseline and follow-up imaging (n = 119), small vessel occlusion (n = 78), anterior/posterior cerebral, basilar artery occlusion (n = 20), and those with critical data unavailable (n = 6). We included 49 patients (Table 1); median age was 70 years old, 55% of them were male and 92% were independent (premorbid mRS 0─1). Sixteen (33%) patients had known or new-onset atrial fibrillation. NIHSS at baseline was a median of 15. Distribution of occlusion site was ICA (n = 26; isolated 10 (20%), tandem 16 (33%)), MCA M1 (n = 14, 29%) and M2 (n = 9, 18%) segment. All thirty-three (67%) patients who received thrombolysis completed treatment before the follow-up MRI. There were no patients with thrombectomy. Time from onset to baseline and to follow-up imaging were a median of 2.0 hours and 20 hours, respectively.

FLAIR image was not available in four patients because of severe motion artifacts. The remaining 45 patients were classified into high (n = 23) or low (n = 22) HVS groups (Table 2). Time from onset to MRI was a median of 20 hours [4.1─26] in the high HVS group, and 23 hours [7.2─26] in the low HVS group. Atrial Fibrillation was present in 11 patients (48%) in the high HVS group, compared to 5 (23%) in the low HVS group. Distribution of occlusion site was: 14 MCA and 9 ICA in the high HVS group, and 9 MCA and 13 ICA in the low HVS group. The high HVS group were more likely to have lower CTPCI (better collateral status) at baseline (median CTPCI in the high HVS group; 28% versus 38%, p = 0.001), smaller core

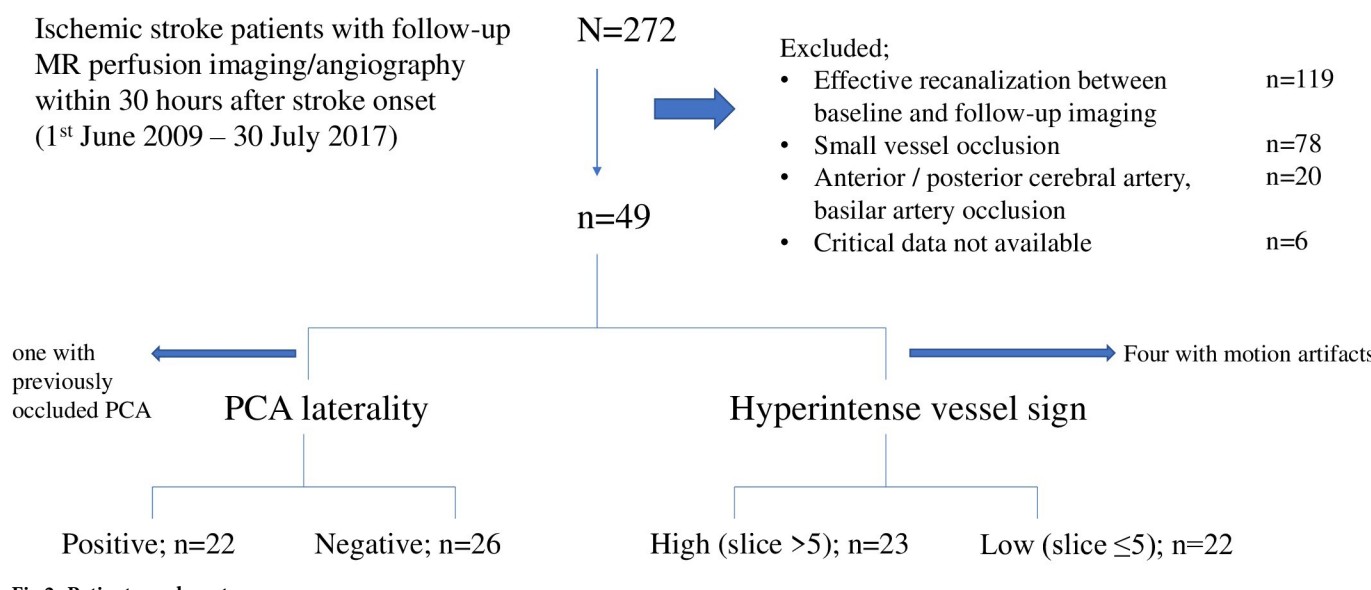

**Fig 2. Patient enrolment.**

volume at baseline (a median of 19 ml versus 58 ml, p<0.001) and follow-up (32 ml versus 109 ml, p = 0.004), more persistent penumbra volume at follow-up (68 ml versus 0 ml, p = 0.001) and more favorable neurological outcome (median mRS at three months; 3 versus 5, p = 0.03).

One patient with a history of previous PCA occlusion was excluded, so 48 patients were classified as PCA laterality positive (n = 22) or negative (n = 26) (Table 2). The positive group

**Table 1. Clinical characteristics of all enrolled patients.**

|  | Total population, N = 49 |
|---|---|
| Age, years old, median, IQR | 70 [62─79] |
| Sex, male, no. (%) | 27 (55) |
| Premorbid mRS 0─1, no. (%) | 45 (92) |
| Risk factors, no. (%) |  |
| Hypertension | 26 (53) |
| Hyperlipidemia | 21 (43) |
| Diabetes mellitus | 9 (18) |
| Atrial fibrillation (known or newly onset) | 16 (33) |
| Ischemic heart disease | 7 (14) |
| Pre-anticoagulant or antiplatelet treatment, no. (%) | 10 (20) |
| NIHSS at baseline, median, IQR | 15 [13─19] |
| Occlusion site, no. (%) |  |
| Internal carotid artery isolated occlusion | 10 (20) |
| tandem occlusion | 16 (33) |
| Middle cerebral artery M1 segment | 14 (29) |
| M2 segment | 9 (18) |
| Thrombolysis, no. (%) | 33 (67) |
| Time from onset to baseline image, hours, median, IQR | 2.0 [1.6─3.2] |
| Time from onset to follow-up image, hours, median, IQR | 20 [4.2─26] |

IQR; interquartile, mRS; modified Rankin Scale, NIHSS; National Institute of Health Stroke Scale

**Table 2. Comparison of clinical characteristics between patients with high or low HVS and with or without PCA laterality.**

|  | HVS | | | PCA laterality | | |
|---|---|---|---|---|---|---|
|  | High, n = 23 | Low, n = 22 | P-value | Positive, n = 22 | Negative, n = 26 | P-value |
| Age, years old, median | 75 [65—79] | 64 [59—71] | 0.06 | 70 [59—72] | 70 [63—79] | 0.29 |
| Sex, male, no. (%) | 11 (48) | 14 (64) | 0.29 | 14 (64) | 13 (50) | 0.34 |
| Premorbid mRS 0—1, no. (%) | 19 (86) | 22 (96) | 0.35 | 21 (95) | 23 (88) | 0.61 |
| Atrial fibrillation, no. (%) | 11 (48) | 5 (23) | 0.08 | 8 (36) | 8 (31) | 0.68 |
| Time from onset to baseline CT, hours, median | 2 [1.6—3.5] | 2.2 [1.6—3.2] | 0.72 | 2.1 [1.6—3.2] | 2 [1.6—2.8] | 0.61 |
| Time from onset to follow-up MRI, hours, median | 20 [4.1—26] | 23 [7.2—26] | 0.51 | 19 [4.2—26] | 20 [4.1—25] | 0.73 |
| NIHSS at baseline, median | 15 | 16 | 0.73 | 15 | 18 | 0.44 |
| at follow-up | 14 | 16 | 0.69 | 18 | 14 | 0.11 |
| MCA occlusion, no. (%) | 14 (61) | 9 (41) | 0.18 | 7 (32) | 15 (58) | 0.07 |
| ICA occlusion, no. (%) | 9 (39) | 13 (59) | - | 15 (68) | 11 (42) | - |
| CTP collateral index at baseline, median, % | 28 [17—32] | 38 [30—54] | 0.001 | 34 [28—42] | 31 [21—44] | 0.59 |
| Baseline perfusion lesion, ml, median | 117 | 139 | 0.61 | 145 | 108 | 0.27 |
| Core volume | 19 | 58 | <0.001 | 39 | 32 | 0.47 |
| Penumbra volume | 98 | 66 | 0.048 | 99 | 74 | 0.14 |
| Follow-up perfusion lesion, ml, median | 143 | 187 | 0.44 | 207 | 135 | 0.34 |
| Core volume | 32 | 109 | 0.004 | 116 | 37 | 0.02 |
| Penumbra volume | 68 | 0 | 0.001 | 34 | 56 | 0.27 |
| Thrombolysis, no. (%) | 17 (74) | 14 (64) | 0.46 | 12 (55) | 21 (81) | 0.05 |
| mRS at three months, median | 3 [2—5] | 5 [3—6] | 0.03 | 5 [3—6] | 4 [3—6] | 0.49 |

CTP; computed tomography perfusion, HVS; hyperintense vessel sign, ICA; internal carotid artery, MCA; middle cerebral artery, mRS; modified Rankin Scale, NIHSS; National Institute of Health Stroke Scale, PCA; posterior cerebral artery

had a higher core volume at follow-up imaging than the negative group (median 116 ml versus 37 ml, p = 0.02) and larger core growth (68 ml versus 4.5 ml, p = 0.02). There were no significant baseline imaging differences between groups, however the rate of ICA occlusion in the PCA laterality positive group was numerically higher, and this approached statistical significance (68%, versus 42% for the negative group; p = 0.07). There was no significant difference in neurological outcome between the two groups. Interestingly, there was a borderline significantly lower rate of thrombolysis in those with PCA laterality positive group (55%, v 81% in the PCA laterality negative group; p = 0.05). Of 22 patients with PCA laterality positive, 12 (55%) had high HVS compared to 10 (38%) in the PCA laterality negative group.

In the multiple regression analysis, PCA laterality positive was related to larger core volume at follow-up, while high HVS was associated with the smaller one. (S1 Table).

## Discussion

We identified 23 patients with prominent HVS on FLAIR image performed a median 20 hours after stroke. The high HVS group showed lower CTPCI suggesting better collateral status. Consistent with this, these patients also had smaller initial core volume, more persistent penumbra at follow-up imaging and more favorable neurological outcomes. We identified 22 patients with positive PCA laterality. Counter to the hypothesis that PCA laterality would be associated with better collaterals and therefore better prognosis, there were significantly larger core growth and larger follow up infarct core volumes in this group, and we did not find other markers of favorable collateral status such as more favorable CTPCI, or better neurological outcome. These findings suggest better performance of high HVS for identification of those

with good collateral status than PCA laterality positivity. This may reflect the fact that PCA-MCA collaterals make a smaller contribution to MCA perfusion than ACA-MCA collaterals [16], however as discussed below, other factors may also have influenced this result.

Prominent HVS within 4.5 hours after stroke has been reported to represent good collateral status and favorable neurological outcomes [10, 11]. We showed similar results in patients with a median 20 hours after stroke onset. The subjects in previous studies were those with MCA (M1 or M2 segment) occlusion, but we found prominent HVS not only in those with isolated MCA occlusion but also in those with tandem or isolated ICA occlusion. ICA occlusion typically leads to whole poor collateral status because of Willisian collateral failure. However, 20% of the total population in the DAWN trial and 37% of those in the DEFUSE 3 trial had ICA occlusion, but showed good collateral status 16–24 hours after stroke [1, 2]. Patients with chronically developed ICA occlusion may be more likely to have better collateral than those with more abrupt occlusion [17, 18]. Such patients may be likely to have prominent HVS even in a late time window.

The clinical implication is that HVS appears to remain useful for interpretation of collateral status, beyond 4.5 hours after stroke onset. Small ischemic core may be difficult to recognize on DWI image and the volume may not be immediately calculated. Prominent HVS suggesting good collateral supply is easy to recognize on FLAIR image, and in addition to clinical-core mismatch, it may assist decision-making to progress to EVT, in hospitals where acute perfusion imaging analysis is not available.

We found larger core growth and final infarct core volumes in the PCA laterality positive group and did not find better CTPCI collateral status or more favorable neurological outcome. There were several possible reasons for this. Whereas subjects in previous studies had MCA M1 segment occlusion [9, 12], we also included patients with ICA occlusion, and ICA was the major occlusion site in the positive group. PCA laterality may be more common in those with ICA occlusion since typically these patients will have poorer ACA-MCA collaterals, due to reduced flow to the ipsilateral ACA. Therefore, there is likely to be lower pressure within the occluded MCA arterial tree, driving higher PCA-MCA collateral flow. However, this confounds the findings regarding PCA laterality since the group with ICA occlusion will do worse. We did not have sufficient numbers of patients to be able to analyse those with and without ICA occlusion separately. A further complicating factor was that the rate of thrombolysis was higher in the group without PCA laterality. Intravenous thrombolysis has previously been reported to reduce core growth rate in patients with large vessel occlusion and non-recanalisation [19], implicating thrombolysis working on distal microvasculature and reducing microvenous thrombosis [20]. Therefore, the higher rate of thrombolysis may also contribute to higher rates of favorable outcome in the PCA laterality negative group. Overall, these factors suggest a high degree of caution in interpretation of the data regarding PCA laterality.

Strengths of this study are that we evaluated collateral status at baseline imaging with CTPCI as continuous variables in each comparison of HVS or PCA laterality. We assessed ischemic core and penumbra volume with baseline and follow-up perfusion imaging, and all perfusion images were analyzed with the same software (MIStar). There are several limitations. The first is that the sample size was unavoidably small, because of the retrospectivity, and strict inclusion criteria. Secondly, thrombolysis may have affected the collateral status on follow-up MRI, although we did exclude patients with effective recanalization between baseline and follow-up imaging to try to avoid this.

## Conclusions

Prominent HVS on FLAIR image in patients with ICA or MCA occlusion up to 30 hours after stroke was associated with good perfusion imaging collateral status, small infarct core volume, persistence of ischaemic penumbra, and favorable clinical outcome. The information of prominent HVS may assist a decision of EVT in a late time window at hospitals where perfusion imaging analysis is not available.

## Supporting information

**S1 Table. Multiple regression analysis of core volume at follow-up image.**
(DOCX)

## Author Contributions

**Conceptualization:** Shinya Tomari, Christopher R. Levi.

**Data curation:** Shinya Tomari, Thomas Lillicrap, Carlos Garcia-Esperon, Longting Lin.

**Formal analysis:** Shinya Tomari.

**Investigation:** Shinya Tomari.

**Methodology:** Shinya Tomari, Yumi Tomari Kashida, Neil J. Spratt.

**Project administration:** Neil J. Spratt.

**Resources:** Shinya Tomari.

**Supervision:** Andrew Bivard, Christopher R. Levi, Neil J. Spratt.

**Validation:** Shinya Tomari, Andrew Bivard, Neil J. Spratt.

**Visualization:** Neil J. Spratt.

**Writing – original draft:** Shinya Tomari.

**Writing – review & editing:** Andrew Bivard, Neil J. Spratt.

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
