## [Decision Letter · Decision Letter 0]

14 May 2024

PONE-D-24-11660Collateral assessment on magnetic resonance imaging/angiography up to 30 hours after stroke onsetPLOS ONE

Dear Dr. Tomari,

Thank you for submitting your manuscript to PLOS ONE. After careful consideration, we feel that it has merit but does not fully meet PLOS ONE’s publication criteria as it currently stands. Therefore, we invite you to submit a revised version of the manuscript that addresses the points raised during the review process.

**3 reviewers have evaluated your manuscript and found important technical issues but also raised interest in your stud. Please address all 3 reviewers' points of critique and provide a point-by-point reply including all necessary modifications of your manuscript.**

We look forward to receiving your revised manuscript.

Kind regards,

Stephan Meckel, MD, PhD

Academic Editor

PLOS ONE

2. The name of the PI specified on the ethical approval document is Mark Parsons. However, Dr. Parsons does not appear as an author on your manuscript. Can you please provide an explanation for why he is not an author?

4. Please ensure that you include a title page within your main document. You should list all authors and all affiliations as per our author instructions and clearly indicate the corresponding author.

Reviewers' comments:

Reviewer's Responses to Questions

**Comments to the Author**

1. Is the manuscript technically sound, and do the data support the conclusions?

Reviewer #1: Yes

Reviewer #2: Yes

Reviewer #3: No

2. Has the statistical analysis been performed appropriately and rigorously? 

Reviewer #1: Yes

Reviewer #2: No

Reviewer #3: Yes

3. Have the authors made all data underlying the findings in their manuscript fully available?

Reviewer #1: Yes

Reviewer #2: Yes

Reviewer #3: Yes

4. Is the manuscript presented in an intelligible fashion and written in standard English?

Reviewer #1: Yes

Reviewer #2: Yes

Reviewer #3: Yes

5. Review Comments to the Author

Reviewer #1: This paper is a valuable report that focuses on two collateral flow indicators, HVS and PCA laterality, and compares them with two perfusion images.

Although relatively old, this dataset is unique because it is now ethically difficult not to perform thrombectomy in patients with acute onset large vessel occlusion. Another feature of the dataset is the relatively large number of tandem occlusions.

It is very interesting in that the classically known HVS is influencing recanalisation therapy decision making, but several areas require further consideration:

Major comments

1. It seems strange that there were no patients in the dataset who had undergone mechanical thrombectomy (MT), as there were already 5RCTs on MT in 2015 when this study was conducted. The inclusion criteria seem to include patients who would be good candidates for MT. Please clarify if any of the excluded patients had undergone MT or if there were any study design or institutional limitations.

2. In this report, it was reported that HVS and PCA laterality were assessed by two readers. Does this mean that two readers evaluated different patients or two readers evaluated the same patients, and if the latter, how were the discrepant cases handled?

3. Prestroke mRS should be reported because of the prognostic impact in the HVS high group and the HVS low group. The aetiology of stroke (e.g. TOAST classification) should also be included as the dataset is characterised by a relatively high number of ICA tandem occlusions. Time from onset to imaging should also be included in the table, as it may be related to HVS and PCA laterality.

Minor comments

1. Please describe the management of patients with M3 or more distal occlusions.

2. The scoring method for HVS seems to be the same as in ref. 11; it is advisable to add this to the METHOD.

Reviewer #2: This is a retrospective study of 49 stroke patients with ICA and/or proximal MCA occlusion with unfavorable recanalization, investigating the relationship between the hyperintense vessel sign (HVS) on FLAIR and PCA laterality on MRA, and their impact on stroke outcomes. The study found that a prominent HVS is associated with a smaller core volume and a larger penumbra volume at follow-up, and better clinical outcomes. In contrast, PCA laterality did not show significant differences in penumbra volumes or outcomes. In fact, it was associated with larger infarct volume at follow-up. The results suggest that HVS is a reliable indicator of good collateral status and favorable prognosis in late-window stroke patients, while PCA laterality does not correlate as strongly with these outcomes.

While the association between collateral status, good functional outcomes, and these radiographic markers was previously reported, those studies only evaluated patients within 4.5 hours from the onset of the stroke. The originality of this study lies in its examination of the same associations within a later time window, between 4.5 and 30 hours after stroke onset.

The clinical significance of these study results lies in the confirmed association between good collateral and penumbral status and the HVS. This finding suggests that HVS could serve as a surrogate marker for perfusion scans. Consequently, HVS could aid in the decision-making process for thrombectomy in patients with large vessel occlusion who are scanned using MRI/MRA but are unable to receive gadolinium contrast for perfusion scans due to medical conditions such as renal failure.

I found one major point of revision that may increase the study’s validity. The authors separated the 49 patients into two groups based on high versus low HVS, and positive versus negative PCA laterality. Since the same patient cohort was categorized using two different criteria, there is likely significant overlap between the groups. For instance, it is plausible that many patients with high HVS also exhibited positive PCA laterality—a correlation previously noted in the literature (PMID 23532013). Therefore, considering the strong correlation between these variables, an adjusted analysis may provide more insight. I recommend conducting an adjusted analysis with the core volume on follow-up scans (and other clinically meaningful aspects) as the dependent variable, incorporating the presence of HVS, PCA laterality, and clot location, especially since the positive PCA laterality group reportedly had a higher incidence of ICA occlusions. This approach would help clarify the independent effects of each variable on patient outcomes, given that the current results are based on unadjusted comparisons.

Here is the suggested minor edit points:

1) Page 2, Line 21, “a late time window up to 30 hours”. I assume the “late time window” means between 4.5h and 30h from the onset of the stroke. Please define the starting point of this “late time window” for clear understanding.

2) Page 2, Line 34, “TIMI grade”. I think TIMI was the typo of modified TICI score. However, if this was legitimate, please provide a reason why the scoring system for coronary revascularization was used to evaluate the result of cerebral thrombectomy.

3) For readers who are not familiar with MIStar, please consider adding a brief description of what Delay Time is and how it differs from Mean Transit Time and Time to Peak.

Reviewer #3: The authors evaluated the association between hypertensive vessel sign on FLAIR imaging (HVS) or PCA laterality on MRA within 30 hours form onset and outcome They showed that prominent HVS in later time was significantly associated with small core, and persistent penumbra volume, and favorable outcome. This study suggested useful imaging markers to predict short term outcome in patients with LVO in anterior circulation presented beyond 4.5 hours; however, there are some questions and comments about contents of the manuscript.

1.When considering the association between HVS and clinical outcome, the possibility of loss of HVS due to the completion of infarct at the same territory should be considered. This study showed the association between high score of HVS and good clinical outcome, but the volume of ischemic core was significantly larger in the low HVS group both at the initial visit and at follow-up. Significant and strong correlation between the low score of HVS and the large ischemic core is not surprising. What is the significance of using HVS rather than initial ischemic core as a predictor of clinical outcome? To elucidate the significance of HVS on clinical outcome, the effect of infarct core volume should be considered.

2.If the follow-up imaging performed within 30 hours after onset (median 20 hours) was used as surrogate for initial assessment of late presenting stroke, as the author mentioned as the limitation of study, the effect of acute treatment, especially intravenous thrombolysis, could not be ignored. Moreover, in this study, the final core volume at the follow-up would correspond to the initial ischemic core in late presenting stroke. If this study was designed for late presenting stroke, true ischemic core measured after follow-up visit should be analyzed. Similarly, clinical baseline parameters including initial NIHSS, were less meaningful. The author should present the NIHSS score obtained at follow-up.

6. PLOS authors have the option to publish the peer review history of their article (what does this mean?). If published, this will include your full peer review and any attached files.

Reviewer #1: No

Reviewer #2: No

Reviewer #3: No

---

## [Author Response · Author response to Decision Letter 0]

27 Jun 2024

We thank the reviewers for their insightful comments, which have helped us significantly improve our manuscript. We hope that our revised manuscript is acceptable for publication.

---

## [Decision Letter · Decision Letter 1]

20 Aug 2024

Collateral assessment on magnetic resonance imaging/angiography up to 30 hours after stroke onset

PONE-D-24-11660R1

Dear Dr. Tomari,

We’re pleased to inform you that your manuscript has been judged scientifically suitable for publication and will be formally accepted for publication once it meets all outstanding technical requirements.

Kind regards,

Stephan Meckel, MD, PhD

Academic Editor

PLOS ONE

Additional Editor Comments (optional):

Reviewers' comments:

Reviewer's Responses to Questions

**Comments to the Author**

1. If the authors have adequately addressed your comments raised in a previous round of review and you feel that this manuscript is now acceptable for publication, you may indicate that here to bypass the “Comments to the Author” section, enter your conflict of interest statement in the “Confidential to Editor” section, and submit your "Accept" recommendation.

Reviewer #1: All comments have been addressed

Reviewer #3: All comments have been addressed

2. Is the manuscript technically sound, and do the data support the conclusions?

Reviewer #1: (No Response)

Reviewer #3: Yes

3. Has the statistical analysis been performed appropriately and rigorously? 

Reviewer #1: (No Response)

Reviewer #3: Yes

4. Have the authors made all data underlying the findings in their manuscript fully available?

Reviewer #1: (No Response)

Reviewer #3: Yes

5. Is the manuscript presented in an intelligible fashion and written in standard English?

Reviewer #1: (No Response)

Reviewer #3: Yes

6. Review Comments to the Author

Reviewer #1: (No Response)

Reviewer #3: The authors have appropriately responded to the reviewer's comment. This manuscript is of satisfactory level for publication in this journal.

7. PLOS authors have the option to publish the peer review history of their article (what does this mean?). If published, this will include your full peer review and any attached files.

Reviewer #1: No

Reviewer #3: No

---

## [Editor Report · Acceptance letter]

22 Aug 2024

PONE-D-24-11660R1 

PLOS ONE

Dear Dr. Tomari, 

I'm pleased to inform you that your manuscript has been deemed suitable for publication in PLOS ONE. Congratulations! Your manuscript is now being handed over to our production team.

Kind regards, 

on behalf of

Prof. Dr. Stephan Meckel 

Academic Editor

PLOS ONE